# Research into the E-Learning Model of Agriculture Technology Companies: Analysis by Deep Learning

**Chi-Hsuan Lin [1], Wei-Chuan Wang [2], Chun-Yung Liu [3], Po-Nien Pan [4,*] and Hou-Ru Pan [5]**

1    College of Business, Minnan Normal University, Quzhou, Fujian 363000, China; lin600115@yahoo.com.tw
2    Department of Banking & Finance, University of Culture, Taipei, Taiwan 111, China; mark1682074@yahoo.com.tw
3    College of Management, Taipei University of Science, Taipei, Taiwan 111, China; perryliu@hotmail.com
4    College of Literature and Media, Yulin Normal University, Yulin, Guangxi 537000, China
5    Department of Banking & Finance, University of Culture, Taipei, Taiwan 111, China; lyr52152@gmail.com
*    Correspondence: wenlly5233@yahoo.com.tw; Tel.: +86-13560901845

**Abstract:** With the advancement of technology, the traditional e-learning model may expand the realm of knowledge and differentiate learning by means of deep learning (DL) and augmented reality (AR) scenarios. These scenarios make use of interactive interfaces that incorporate various operating methods, angles, perceptions, and experiences, and also draw on multimedia content and active interactive models. Modern education emphasizes that learning should occur in the process of constructing knowledge scenarios and should proceed through learning scenarios and activities. Compared to traditional "spoon-feeding" education, the model learning scenario is initiated with the learner at the center, allowing the person involved in the learning activity to solve problems and further develop their individual capabilities through exploring, thinking and a series of interactions and feedback. This study examined how students in the agriculture technological industry make use of AR digital learning to develop their industry-related knowledge and techniques to become stronger and more mature so that they unconsciously apply these techniques as employees, as well as encouraging innovative thought and methods to create new value for the enterprise.

**Keywords:** augmented reality (AR); deep learning (DL); agriculture technology

## 1. Introduction

Deep learning mainly involves a neural network with large hidden layers for extremely difficult machine learning tasks such as 3D object recognition, voice recognition, and so on. Data mining includes all techniques, including deep learning, that can help one make sense of data.

The objective of this study was to explore the key success factors for employees of agriculture technology companies in accepting augmented reality (AR) e-learning from the deep learning (DL) point of view.

E-learning is already widely recognized for its advantages, namely that it is not restricted by time and space and is low cost. The main purpose of e-learning is that the teacher conveys their individual knowledge and expertise or practice test questions to educate students and train users. This learning model is mostly successful in teaching and explaining set theory, or fortifies memory training to allow trainees to familiarize themselves with the lesson content. With regard to the enhancement of the user's thinking ability, logical reasoning or man and machine interaction, etc., the effects of e-learning are quite slim. In recent years, since the internet and mobile technology have become widespread, a massive quantity of data (so-called big data) together with data mining technology have combined various pieces of small data and then disseminated them. The correlations between these numerous

pieces of information and data turn them into an understandable structure which has applications for industry [1].

The rise of deep learning has allowed researchers to discover the potential power of DL as a supporting tool. If extended to agriculture technology companies, this may not only take the place of the traditional manual handling of activities by the staff—such as foreign exchange, gold trading, lending, borrowing and stock exchange—but may also utilize automated systems such as apps and robo-advisors to further care for customers' wealth and investment planning.

The rise of agriculture technology companies in recent years is one area that has been widely noted in the new economic industry and employees of this highly competitive and growing industry may seek to acquire knowledge through DL, in order to use machine learning characteristics to solve more issues, reduce mistakes and further enhance agricultural service quality and efficiency in order to create new value for the industry.

This first chapter introduced the background and purpose of the research, the second chapter reviews the literature, the third chapter describes the research methods, the fourth chapter presents the research results, and the fifth chapter provides the conclusions and suggestions.

## 2. Literature Review

The theoretical basis of this study and the framework for the development of the research combine the technical applications of e-learning with augmented reality (AR) to specifically investigate agriculture technology companies. The study also looks at different theoretical research into deep learning (DL) technologies. A literature review was conducted before further integration was undertaken and the relationships between the three theories were analyzed.

### 2.1. E-Learning

Motiwalla has pointed out that in the past, e-learning usually referred to a person who learned in front of a computer using an e-learning lesson [2]. After the advancement and popularity of technology and mobile networks, mobile learning (m-learning) has become a popular learning format for students in the current generation. Lehner and Nosekabel defined m-learning as providing digital teaching materials and information via services and equipment without restrictions of time and place. All learning that is covered by this definition is m-learning [3]. M-learning is a genuine way to provide an environment that makes information available on hand and makes learning possible anytime and anywhere. Simply put, it is possible to learn anything, anytime and anywhere by fully utilizing mobile devices and the network environment, as well as realizing a variety of learning opportunities [4].

E-learning in the network environment is centered on the teacher, who uses streaming to convey the teaching materials and the content. The development of webpage technology is applied in e-Learning. This has developed from e-learning 1.0, which mainly provided web-based courseware, to e-learning 2.0, which combines social media and other sites such as YouTube, blogs, Facebook and Wikipedia as well as cloud networks to provide students with a good interactive and co-working environment [5].

The use of modern information techniques may place the focus on the individual learner to find teaching materials of various kinds to enhance the teaching effect, which is one of the main objectives of e-learning. M-learning equipment includes MP3 players, portable devices, e-book readers, tablets, mobile phones, augmented reality, and other wearable technologies. This study will enquire into the learning effect of augmented reality as part of e-learning, and understand the factors that affect acceptance and influence employees at agriculture technology companies from the learner's point of view, in order to achieve the goal of learning by further deepening their professional abilities.

### 2.2. Augmented Reality

Augmented reality (AR) means combining virtual information such as pictures, videos, or texts with the image actually seen in the real world through a monitor to form a compound image.

This compound image emphasizes the interaction between the real world and the virtual world to allow the learner to engage with a completely different learning model to aid with the acquisition of knowledge and information. Many people often mistakenly believe that augmented reality is the same as virtual reality (VR). AR emphasizes the combination of virtual objects and reality to present objects realistically to the eye, but VR uses computers to simulate and completely create a three-dimension virtual world to provide the user with the visual impression that they are looking at a real scene and allowing the user to be immersed as if in the real world, being able to observe realistic objects in three dimensions in real time without restriction [6].

Augmented reality has two current definitions. The first definition is Milgram's reality–virtuality continuum proposed in 1994 by Milgram, Takemura, Utsumi and Kishino, et al. [7]. They placed the virtual environment and the real environment at separate ends of a continuum and called the space in between mixed reality. The area closer to the real environment was called augmented reality and the area closer to the virtual environment was called augmented virtuality, as can be seen in Figure 1.

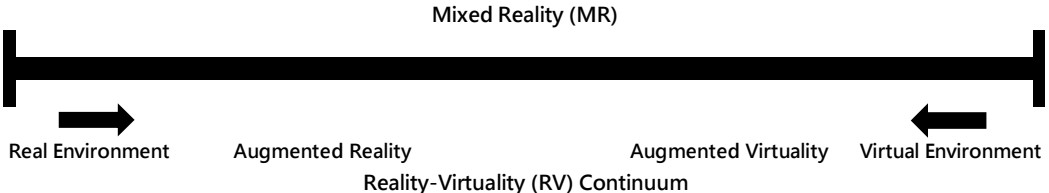

**Figure 1.** Reality–Virtuality (RV) Continuum. Source: Milgram, et al., (1994) SPIE. Kyoto, Japan: 283 [7].

The second definition is from Azuma [6], who considers AR to include three aspects: (1) combines virtual objects with reality; (2) interacts in real time; and (3) is three-dimensional.

By means of positioning, image recognition and program development techniques, a user may review digital data—such as texts, pictures, sounds and videos—superimposed on real world visuals on mobile devices such as smart phones and tablet computers [8]. There are varying methods of integration, and viewing can be arranged through direct 3D projection for space augmentation, and through monitoring equipment, video cameras or helmet-mounted displays (HMDs). In 2015, Microsoft released HoloLens, the first HMD that was capable of auto-projecting augmented reality, and in 2017 they presented a conceptual product, the holographic near-eye display, which can satisfy the requirements of AR and VR applications, but are not as heavy and uncomfortable as the traditional displays [9]. Augmented reality (AR), in particular that from Affordance technology, is very suitable for situated learning and teaching for cognitive apprenticeships [10], as well as training for procedural tasks and problem-based learning models.

*2.3. Deep Learning*

The term deep learning (DL) was first presented in 1956 at the Dartmouth conference at Dartmouth College, which sought to define deep learning (to equip a machine with the ability to imitate a human being). This historical conference achieved a major advancement in deep learning.

Human–machine intelligence, instead of hard-coded control mechanisms, can be used to defend against various attacks, especially against complex attacks. The most important manifestation of this intelligence is machine learning and deep learning; adding e-learning will even further strengthen the effect [11].

Currently, deep learning has substantial commercial value and many successful application cases. Examples of deep learning applications in the agricultural services are as follows:

2.3.1. Agricultural Trading and Personal Finance Consultation

Robo-advisor is a robot with deep learning as the core, supported with complicated software but transformed into a simple website interface, guiding customers according to the different

objectives and needs of investors into suitable investment combinations and various asset management planning scenarios.

### 2.3.2. Identification

Biometric technology can be used as a primary means of identification for customers in agricultural trading and for personal protection on a designated site. Biometric technology and sensor devices are used for biological traits such as face, voice, voice print, iris, vein, fingerprint, etc. These identification operations would in the past have required clients to be at a physical branch of the bank to conduct face-to-face identification, but may in the future be carried out with mobile devices (such as mobile phones and tablets) and computers for long-distance identification, which will immensely enhance efficiency and reduce cost.

### 2.3.3. Smart Customer Service

At present, many banks make use of Pepper robots as guest relation officers. In addition to the primary service of saying hello, gaming and providing and checking information, they may utilize deep learning to immediately certify customer identification by means of facial recognition functions. They may also use big data and search engines to provide instant banking information and customized banking products with more precise marketing information.

### 2.3.4. Regulation Technology (RegTech)

Deep learning can also be used in regulation technology (RegTech)-related management. In order to deal with the rise of agricultural technology and thus a more and more complicated market environment, agricultural regulation and legal adherence control, Deutsche Bank has used deep learning technology to regularly filter and inspect videos of the dialogue between bank staff and customers or recorded information using designated key words to speedily and effectively clarify whether they are in violation of the relevant operational regulations.

### 2.3.5. Precision Marketing

Big data may analyze the traits of customer and social group activity and behavior to provide differentiated products and services. The core product and service of the agricultural industry is credit (loan limits), price setting (bank rates and fees) and wealth management.

### *2.4. Information System Success Model*

Information system success has been studied since the 1980s and has been explored by many scholars and specialists, of whom DeLone and McLean are the best representatives [12]. After collecting a large quantity of literature and in-depth study and research, they presented the information system success model (I/S success model) in 1992.

More than a decade later, DeLone and McLean [12] further presented a revised version of the I/S success model. In addition to further explanation of the original theoretical framework, they followed Pitt, Watson and Kavan's (1995) suggestion to include service quality, and also added a new facet, net benefit [13]. They based the new model on the feedback relation between use, user satisfaction and net benefit to evaluate the system's effect and considered it a success index. This extended the scope of the model to various e-commerce fields, where it is generally called the "D&M Information System Success Model" [13]. The revised model is shown in Figure 2.

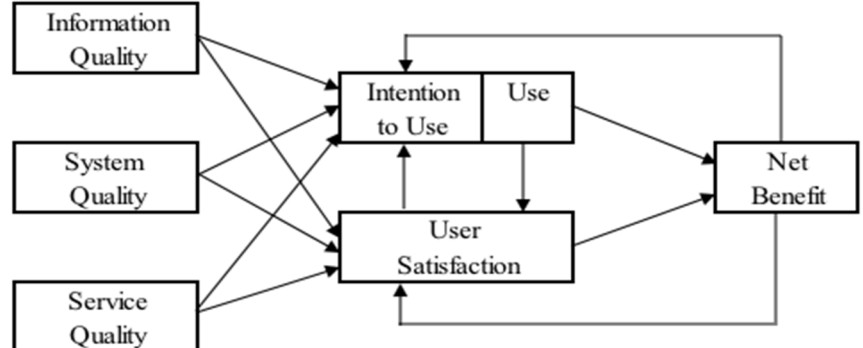

**Figure 2.** D&M information system success model (D&M I/S success model). Source: W. H. DeLone and E. R. McLean [14], Journal of Management Information System.

The model after revisions included six perspectives:

1.  Information quality: the output quality, including completeness, correctness, clarity, negotiability, intelligibility, reliability, utility, compactness, reliability, objectivity, immediacy, and novelty.
2.  System quality: information system quality, including easy operability, acquirability, functional usefulness, correctness, integration and flexibility, and efficiency.
3.  Service quality: the ability of suppliers of an information system to adjust services, support systems, etc.
4.  Intention of use: the intensity of the user's initiative in using the information system, i.e., the individual's subjective willingness.
5.  User satisfaction: the user's degree of satisfaction, this is a general index including soft/hardware, the system interface and satisfaction with policy decisions.
6.  Net benefit: the benefit that the information system may bring, including invisible and visible benefits. The same information system may present different benefits to different organizations.

If an e-learning user has efficiently achieved the learning objective, the information quality and system quality aspects of the D&M I/S success model will have had a positive influence on the satisfaction of the user and the information quality will also have had a positive influence on the use [14].

*2.5. Expectation Confirmation Theory (ECT)*

The expectation confirmation theory is mainly used to assess and evaluate consumer satisfaction with a service or product as well as the behavior after purchase [15], as the basic framework for the research model of customer satisfaction.

Expectation confirmation theory, which showed in Figure 3, was presented by Oliver, and the concept is: (1) a consumer will hold a certain degree of expectations regarding a certain product or service before purchase, (2) after the consumer experiences the product or service for some time they will have formed an opinion of the perceived performance and will form a new opinion of the product or service, (3) the consumer will make a comparison between the actual performance after their experience of the product (or service) and the initial expectation in order to assess and confirm the confirmation of expectation, and (4) the result of this comparison will determine the degree of satisfaction, which will affect the possibility of a repurchase intention [15].

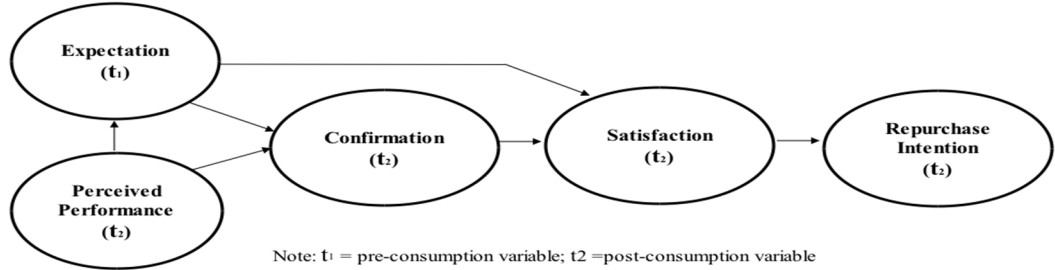

**Figure 3.** Expectation confirmation theory (ECT) [15].

Therefore, expectation confirmation theory mainly illustrates the consumer repurchase intention and explains the consumer expectation before the purchase of a product (or service) and the difference to the reaction after the purchase. This reaction is called confirmation. Bhattacherjee integrated ECT and the technology acceptance model (TAM) and presented the expectation confirmation model (ECM), which is a post-acceptance model of IS continuance [16].

Confirmation is a decision based on the integration of expectation and performance, and may be an important factor influencing user satisfaction. It can therefore be inferred that both expectation and performance will have a positive effect on user satisfaction and that confirmation has the same positive effect as the sense of previous use, which will be discussed below.

*2.6. Theory of Communicative Action*

The theory of communicative action was developed by Habermas based on rationality. According to the theory of communicative action, in order to achieve a rational society, it is necessary to reconstruct the communicative ability of human beings through inspiration, reflection and criticism to achieve communicative purposes that are independent, mature and free. The basic content of the communicative action theory includes:

1. Communicative competence and communication speech action: Communicative competence is the basis and objective of Habermas' theory. It involves the speaker's individual grasp of the scenario in the internal, social and natural fields, requiring a suitable connection to achieve successful communication. Habermas has pointed out that speech and action include a propositional component are illocutionary.

The propositional component is the content that the speaker wants to convey; while the illocutionary component is the submerged expression, gesture and intonation of the speaker. Therefore, communication requires both these two points to be understood by the listener to be considered a successful speech interaction.

2. Claim of the effectiveness of speech: Habermas pointed out that speech is a communication of mutual understanding, not only speaking grammatical sentences and words, but more importantly building a relationship that both parties mutually approve. Chiang pointed out in their study that Habermas' theory of communicative action may be divided into four levels and turned into a validity claim [17]:

i.   Rightness claim: The speaker's content conforms to common specifications.
ii.  Truth claim: The facts of the statement are acceptable to the listener.
iii. Truthfulness claim: The speaker is very sincere in obtaining the trust of the listener.
iv.  Comprehensibility claim: The content of the speaker conforms to grammar.

3. Ideal communicative speech scenario: Of the above four claims, the truthfulness claim and the comprehensibility claim are easy to accept, but when the rightness claim and the truth claim are questioned, a rational solution must be sought. This is called discursive discourse. Wakefield and Warren considered that exchange behavior provides teachers and students with a meaningful communicative opportunity to obtain or construct a comprehension objective. Nah and Chung pointed out in their research based on the theory of communicative action that the media in

modern society not only plays a delivery role, but through the internet in this digital age, the media also creates a mutual, multiple correlation realm between reporters and citizens, who can conduct interactions and discussions in this space [18].

This study applied the theory of communicative action to explore the positive influence, satisfaction and intention when staff from the agricultural industry participate in e-learning on rational discussion and communicative behavior. Expectation confirmation theory (ECT) was mainly applied to assess and evaluate customer satisfaction and post-purchase behavior [15], serving as the basic study framework for the consumer satisfaction scale.

## 3. Methodology

### 3.1. Study Framework

AR applications that use deep learning need to be educated using an object with information shown on it. For example, when the source object is a human hand, recognition can be performed for all of the different positions of any human hand. It is clear that deep learning will be effective in the future of AR. The main purpose of using AR to date has been to enhance students' interest in e-learning as well as providing them with additional information. AR educational games and AR for lab experiments are also growing fields. This study explored deep learning using the information system success model, and then studied the effect of e-learning with expectation confirmation theory. Finally, the effect of AR was confirmed using the communicative action theory.

This study used three theoretical frameworks—the information system success model (I/S success model), expectation confirmation theory (ECT) and the theory of communicative action—and the theory of reasoned action (TRA) to forecast individual behavior in order to explore the factors that enable agricultural industry employees to successfully accept e-learning and to internalize the material in order to improve their skills and apply them subconsciously. This paper will further present innovative ideas and practices for the industry.

The above theory is presented in the diagram in Figure 4.

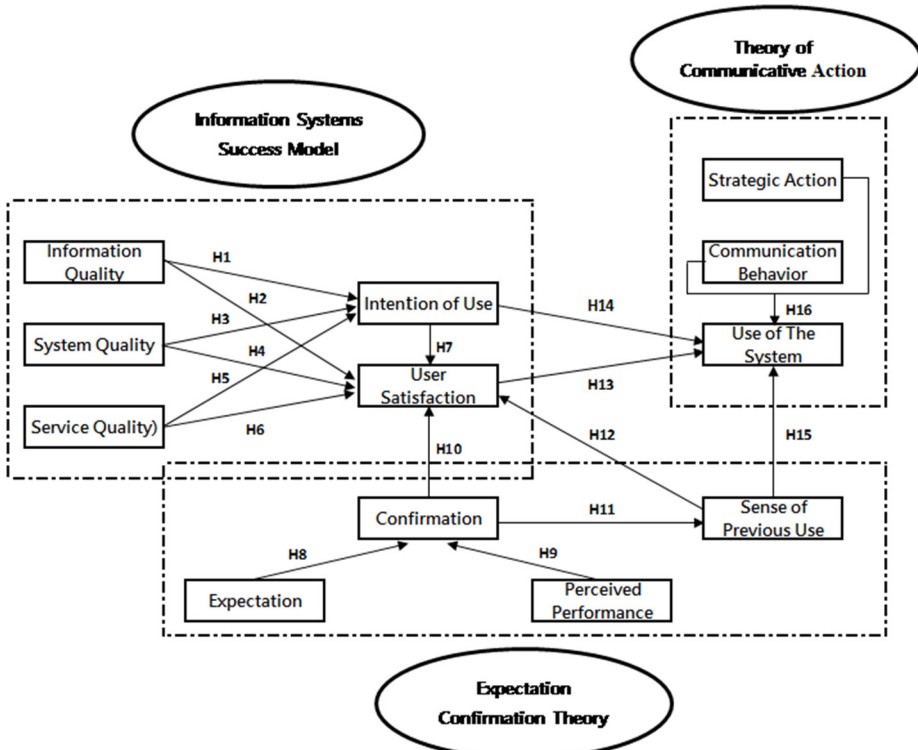

**Figure 4.** Study framework. Source: Authors' projection in Partial Least Squares (PLS).

The following assumptions are inferred:

**Hypothesis 1** (H1): *Information quality positively affects the user's intention of use of e-learning.*

**Hypothesis 2** (H2): *Information quality positively affects user satisfaction after e-learning.*

**Hypothesis 3** (H3): *System quality positively affects the user's intention of use of e-learning.*

**Hypothesis 4** (H4): *System quality positively affects user satisfaction after e-learning.*

**Hypothesis 5** (H5): *Service quality positively affects the user's intention of use of e-learning.*

**Hypothesis 6** (H6): *Service quality positively affects user satisfaction after e-learning.*

**Hypothesis 7** (H7): *The intention of use positively affects user satisfaction after e-learning.*

**Hypothesis 8** (H8): *The user's expectation positively affects confirmation after e-learning.*

**Hypothesis 9** (H9): *The user's perceived performance positively affects confirmation after e-learning.*

**Hypothesis 10** (H10): *The user's confirmation positively affects user satisfaction after e-learning.*

**Hypothesis 11** (H11): *The user's confirmation positively affects the sense of previous use after e-learning.*

**Hypothesis 12** (H12): *The user's sense of previous use positively affects user satisfaction.*

**Hypothesis 13** (H13): *The user's user satisfaction positively affects the use of the system.*

**Hypothesis 14** (H14): *The user's intention of use positively affects the use of the system.*

**Hypothesis 15** (H15): *The user's sense of previous use positively affects the use of the system.*

**Hypothesis 16** (H16): *Strategic action and communication behavior positively affects the use of the system.*

*3.2. Study Object and Questionnaire Design*

This study and the study questionnaire were designed based on the questionnaires of the three theoretical models—the information system success model, expectation confirmation theory, and the theory of communicative action—combining them with the characteristics of AR and deep learning, so that the questionnaire design was logical and had structural validity in order to facilitate statistics and quantification. The survey questionnaire covered three categories of respondent:

1. People working in agriculture-related business and related scholars and students.
2. People working in information and communication-related businesses and related scholars and students.
3. People working in fields not covered by the above and related scholars and students.

To ensure the efficiency of the sample collection, this study issues 500 questionnaires through normal distribution. The questionnaires were valid from November 2017 to February 2018. The questionnaire was designed specifically to investigate the information quality, system quality, service quality, intention of use, user satisfaction, and the use of the system. The questionnaire utilized the Likert Scale for measurements.

In the questionnaire design, especially the design of the AR and e-learning questionnaire, the questionnaire project was derived from the theory of communicative action and expectation confirmation theory. Thus, the AR questionnaire project had two major categories: strategic action

and communication behavior, and the e-learning questionnaire covered expectation, confirmation, perceived performance, and sense of previous use.

## 3.3. Study Analysis Method

This study adopted structural equation modeling (SEM), combining multiple regression and factor analysis, and mainly looked at potential variable items to proceed with the route analysis and examine the cause and effect route relations for potential variable items. The study adopted the PLS 2.0 (PLS-SEM) tool to produce the structure equation model.

The structural equation model is a statistics model with a multilinear model to present analytical variation relation that mainly analyzes using invisible potential variable factors and enquires into the joint relations among variables. The structural equation model has two main purposes: one is to establish a statistical model of high fit based on the logic relation of multiple variables, and the other is to demonstrate a system relation through the structural coefficient to produce strategic implications and the logical relation of variables. The framework of the structural equation theory includes the structural model and measurement model.

## 4. Study Results and Discussion

### 4.1. Basic Information Analysis

For the purposes of this study, a total of 463 effective questionnaires were obtained. Of these, 261 respondents were male (56%) and 202 were female (44%). With regard to age, the majority of respondents—167 people (36.07%)—were over 40 years of age, and the second largest group of respondents were 25 to 30 years of age, with 105 respondents (22.68%). The frequency of use was based on the respondents' own practical experience. With regard to the frequency of use of e-learning, the majority of respondents reported nearly no use—152 respondents (32.83%)—and 149 respondents (32.18%) reported using e-learning once a week. In terms of their experience of e-learning, 369 respondents (79.70%) had taken less than 5 classes, and 78 respondents (16.85%) had taken 6–10 classes. With regard to the satisfaction with e-Learning, 299 respondents (64.58%) rated their experience as satisfactory and 131 respondents (28.29%) rated their experience as very satisfactory. The basic information of the sample is outlined in Table 1:

**Table 1.** Basic information of the sample.

| Basic Data Items. | Category | Number of People | Percentage | Accumulated Percentage |
|---|---|---|---|---|
| Gender | Male | 261 | 56% | 56% |
| | Female | 202 | 44% | 100% |
| Business | Network business | 38 | 8.21% | 8.21% |
| | Telecommunications business | 41 | 8.86% | 17.06% |
| | Electronic spareparts | 49 | 10.58% | 27.65% |
| | Electronic engineering department | 50 | 10.80% | 38.44% |
| | Banking business | 41 | 8.86% | 47.30% |
| | Insurance business | 43 | 9.29% | 56.59% |
| | Investment credit business | 38 | 8.21% | 64.79% |
| | Agricultural department | 55 | 11.88% | 76.67% |
| | Restaurant tourism business | 20 | 4.32% | 80.99% |
| | Mechanical car and motorbike business | 35 | 7.56% | 88.55% |
| | Military, government and teaching business | 40 | 8.64% | 97.19% |
| | Others | 13 | 2.81% | 100% |
| Age | 20–25 | 105 | 22.68% | 22.68% |
| | 25-30 | 92 | 19.87% | 42.55% |
| | 30-35 | 99 | 21.38% | 63.93% |
| | 40+ | 167 | 36.07% | 100% |
| Frequency | Nearly daily | 18 | 3.89% | 3.89% |
| | Once in three days | 55 | 11.88% | 15.77% |
| | Once a week | 149 | 32.18% | 47.95% |
| | Once a month | 89 | 19.22% | 67.17% |
| | Almost never | 152 | 32.83% | 100% |

| Basic Data Items. | Category | Number of People | Percentage | Accumulated Percentage |
|---|---|---|---|---|
| Experience | Less than 5 classes (inclusive) | 369 | 79.70% | 79.70% |
| | 6–10 classes | 78 | 16.85% | 96.54% |
| | 11–20 classes | 12 | 2.59% | 99.14% |
| | 21 classes and over | 4 | 0.86% | 100% |
| Degree of satisfaction | Very unsatisfied | 6 | 1.30% | 1.30% |
| | Unsatisfied | 14 | 3.02% | 4.32% |
| | Satisfied | 299 | 64.58% | 68.90% |
| | Very satisfied | 131 | 28.29% | 97.19% |
| | Extremely satisfied | 13 | 2.81% | 100% |

Source: Author's projection using SPSS.

## 4.2. Measurement Model Analysis

In order to further examine the reliability and efficiency of the measurement questionnaire, this study used the PLS statistics software. The examination of the measurement model included internal consistency, convergent validity and discriminating validity.

The purpose of reliability analysis is to understand measurement consistency, stability and internal consistency. The higher the reliability, the more stable the measurement.

Composite reliability (CR) is the consistency of the facet internal variable. If the potential variable CR value is high, it means that the measurement variable is related to height, and that it is therefore easier to measure the potential variable. In general, the composite reliability must be higher than 0.7. Table 2 shows that the CR of all facets in this study was higher than the threshold value of 0.7, demonstrating that the internal consistency of the measurement tool designed by this study was acceptable.

**Table 2.** Total results of the measurement model.

| | AVE | CR Value | R Square | Cronbach's Alpha |
|---|---|---|---|---|
| Information Quality | 0.841 | 0.941 | 0.000 | 0.905 |
| System Quality | 0.724 | 0.887 | 0.000 | 0.809 |
| Service Quality | 0.727 | 0.889 | 0.000 | 0.811 |
| Intention of Use | 0.677 | 0.913 | 0.728 | 0.880 |
| User Satisfaction | 0.642 | 0.900 | 0.632 | 0.860 |
| Expectation | 0.649 | 0.902 | 0.000 | 0.864 |
| Perceived Performance | 0.685 | 0.915 | 0.000 | 0.884 |
| Confirmation | 0.699 | 0.903 | 0.588 | 0.856 |
| Strategic Action | 0.696 | 0.901 | 0.000 | 0.854 |
| Communication Behavior | 0.666 | 0.888 | 0.000 | 0.831 |
| Use of The System | 0.722 | 0.886 | 0.647 | 0.808 |
| Sense of Previous Use | 0.656 | 0.905 | 0.536 | 0.869 |

Source: Author's projection in PLS.

The average variance extracted (AVE) of the convergent validity extracted for an individual facet must at least be greater than 0.5, i.e., the facet must have sufficient convergent validity [19]. Table 2 shows that the AVE value of the facets in this study were greater than 0.5, meaning that the measurement model has very good convergent validity.

To obtain the discriminating validity, the relative degree of reliability of every variable and the measurement of every other variable of the same facet should be higher than the measurement of the relative coefficient of variables from different facets. The discriminating validity of the average variance extracted (AVE) of the individual facet square root (Table 3 diagonal value) should be greater than the relative coefficient between the facet and others facet in the model (Table 3 non-diagonal value) [20]. As shown in Table 3, the diagonal square root of the average variance extracted (AVE) was greater than the coefficient between any two facets, demonstrating the actual variance of the variables in the measurement model; in other words, the measurement model of this study had very good distinguishing validity.

**Table 3.** All potential variable relations.

| | Average | Standard Difference | 1 | 2 | 3 | 4 | 5 | 6 | 7 | 8 | 9 | 10 | 11 | 12 |
|---|---|---|---|---|---|---|---|---|---|---|---|---|---|---|
| Information Quality (1) | 3.622 | 0.575 | 0.917 | | | | | | | | | | | |
| System Quality (2) | 3.538 | 0.596 | 0.673 ** | 0.851 | | | | | | | | | | |
| Service Quality (3) | 3.509 | 0.669 | 0.521 ** | 0.565 ** | 0.853 | | | | | | | | | |
| Intention of Use (4) | 3.442 | 0.633 | 0.572 ** | 0.577 ** | 0.524 ** | 0.823 | | | | | | | | |
| User Satisfaction (5) | 3.421 | 0.68 | 0.446 ** | 0.439 ** | 0.497 ** | 0.597 ** | 0.801 | | | | | | | |
| Expectation (6) | 3.348 | 0.706 | 0.440 ** | 0.507 ** | 0.464 ** | 0.551 ** | 0.597 ** | 0.806 | | | | | | |
| Perceived Performance (7) | 3.557 | 0.612 | 0.553 ** | 0.524 ** | 0.479 ** | 0.595 ** | 0.478 | 0.479 ** | 0.828 | | | | | |
| Confirmation (8) | 3.642 | 0.621 | 0.616 ** | 0.594 ** | 0.517 ** | 0.584 ** | 0.481 | 0.469 ** | 0.664 ** | 0.836 | | | | |
| Strategic Action (9) | 3.541 | 0.591 | 0.591 ** | 0.617 ** | 0.490 ** | 0.517 ** | 0.46 | 0.508 ** | 0.561 | 0.656 ** | 0.834 | | | |
| Communication Behavior (10) | 3.438 | 0.612 | 0.542 ** | 0.563 ** | 0.520 ** | 0.583 ** | 0.582 | 0.606 ** | 0.555 | 0.630 ** | 0.703 ** | 0.816 | | |
| Use of The System (11) | 3.561 | 0.6 | 0.640 ** | 0.647 ** | 0.522 ** | 0.544 ** | 0.523 | 0.482 ** | 0.584 | 0.617 ** | 0.663 | 0.653 ** | 0.850 | |
| Sense of Previous Use (12) | 3.474 | 0.605 | 0.591 ** | 0.604 ** | 0.485 ** | 0.509 ** | 0.52 | 0.557 ** | 0.554 | 0.603 ** | 0.667 | 0.667 ** | 0.720 ** | 0.810 |

Source: Author's projection in PLS; Remarks: 1. The value of the diagonal line is the square root value, and the non-diagonal line is the relative coefficient among variables. 2. ** $p < 0.01$; 3. The square root of the AVE for an individual facet should be greater than the relative coefficient between the facet and other facets in the model, indicating the distinguishing validity.

### 4.3. Structure Model Analysis and Qualification

After confirmation of the reliability and validity, the facets reached a certain level in the measurement model. This study then implemented a structure model analysis and qualification of the causal relations among potential variables using SmartPLS and used R2 to evaluate the explanatory ability. R2 refers to the percentage of variance that the external variable can explain of the internal variable and represents the forecast ability of the study model. The value of R2 is between 0 and 1, and the greater the value, the better the explanatory ability. It should be noted that the route qualification results show that not only were information quality, system quality and service quality positively affected by intention to use and user satisfaction, but the sense of previous use also had a substantial effect on the use of the system, as seen in Figure 5.

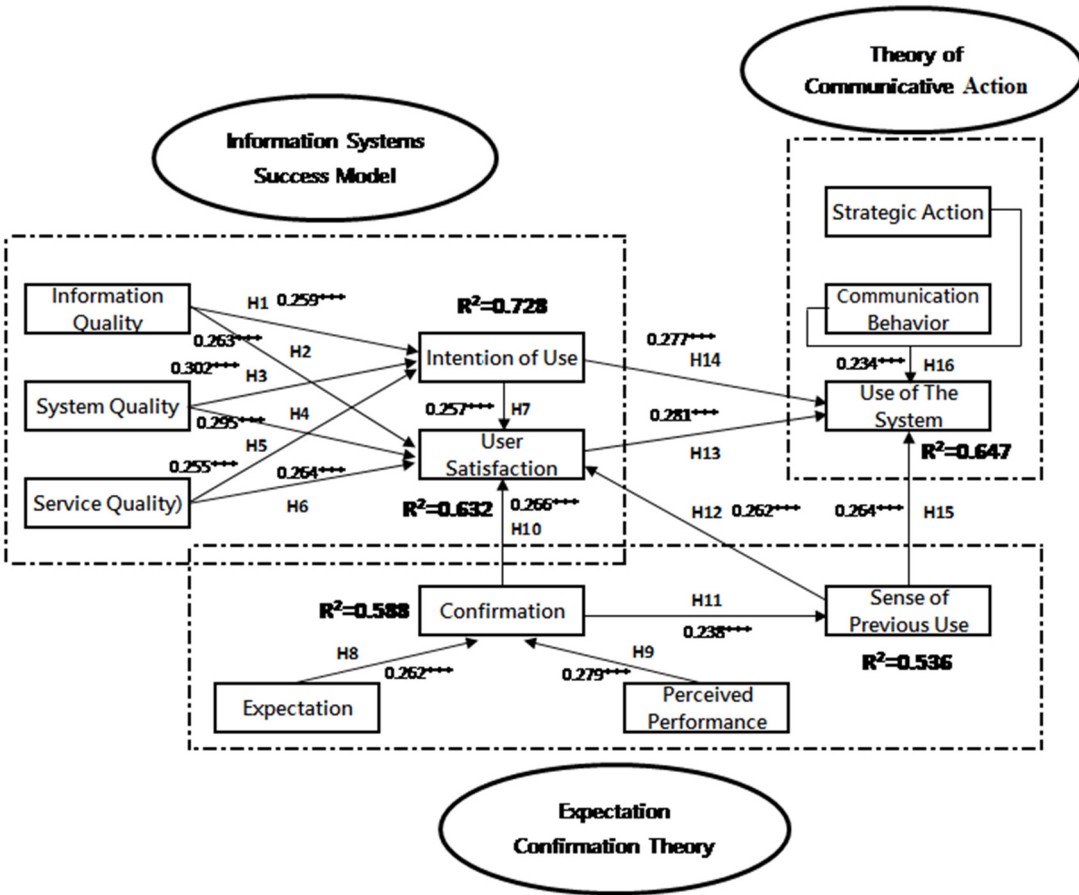

**Figure 5.** Structural model relation route qualification results. Source: Author's projection in PLS.

The study assumption qualification results are shown in Table 4.

**Table 4.** Structural model study assumption and assumption qualification results.

| Assumption | Relation | Path Coefficient | T Value | Significance | Verification Result |
|---|---|---|---|---|---|
| H1: Information Quality → Intention of Use | + | 0.259 | 3.953 | *** | Valid |
| H2: Information Quality → User Satisfaction | + | 0.263 | 6.521 | *** | Valid |
| H3: System Quality → Intention of Use | + | 0.302 | 7.209 | *** | Valid |
| H4: System Quality → User Satisfaction | + | 0.295 | 6.752 | *** | Valid |
| H5: Service Quality → Intention of Use | + | 0.255 | 2.717 | *** | Valid |
| H6: Service Quality → User Satisfaction | + | 0.264 | 2.689 | ** | Valid |
| H7: Intention of Use → User Satisfaction | + | 0.257 | 1.113 | — | Invalid |
| H8: Expectation → Confirmation | + | 0.262 | 1.433 | — | Invalid |
| H9: Perceived Performance → Confirmation | + | 0.279 | 1.679 | * | Valid |
| H10: Confirmation → User Satisfaction | + | 0.266 | 1.704 | * | Valid |
| H11: Confirmation → Sense of Previous Use | + | 0.238 | 0.968 | — | Invalid |
| H12: Sense of Previous Use → User Satisfaction | + | 0.262 | 1.063 | — | Invalid |
| H13: User Satisfaction → Use of the System | + | 0.281 | 7.968 | *** | Valid |
| H14: Intention of Use → Use of the System | + | 0.277 | 1.875 | * | Valid |
| H15: Sense of Previous Use → Use of the system | + | 0.264 | 0.896 | — | Invalid |
| H16: Communication Behavior → Use of the System | + | 0.234 | 1.934 | ** | Valid |

Significant level: * $p < 0.05$, ** $p < 0.01$, *** $p < 0.001$. Source: Author's projection in PLS.

Based on the above analysis, the study findings are summarized as follows:

1.  In the information systems success model, information quality, system quality and service quality positively affected intention of use and user satisfaction. The assumptions of H1–H6 were established. System quality most strongly affected intention of use ($p < 0.001$), followed by user satisfaction ($p < 0.001$).

2.  In the expectation–confirmation model, perceived performance had a positive effect on confirmation, and confirmation also had a positive effect on user satisfaction. Thus, both H9 and H10 were established.

3.  In the communicative action model, intention of use, user satisfaction and communication behavior positively affected use of the system, and thus assumptions H13, H14 and H16 were all established, certifying the reliability and validity of the measurement questionnaire and the suitability of adopting the PLS statistics software for the examination of the measurement model. This examination included internal consistency, convergent validity and discriminate validity.

## 5. Conclusion and Suggestions

### 5.1. Conclusion

The use of PLS 2.0 (PLS-SEM) helped to establish the connected key factors that affect employees of the agricultural industry in terms of enhancing their work efficiency through AR e-learning. Various facets and evaluation criteria were evaluated, and percentage values and preferences were established, including for the aspects: information system, expectation confirmation and communicative action. Three major models and 12-facet evaluation criteria were used. The models ranked in order from the most to least important are: information system, expectation confirmation and communicative action.

### 5.2. Study Contribution

1. AR e-learning using DL is a successful model.

In the opinion of scholars and experts, all industries are depending more and more on deep learning, and the success of e-learning is affected by the vital fact of whether or not DL is applied. DL algorithms are capable of rapidly searching for the most suitable answer in a broad information space, and make up for the time waste and error caused by mass network information, thus assisting the user to quickly and effectively obtain suitable learning information for him/herself.

The development of deep learning will allow future education to evolve into individual problem solving that is problem-oriented, rather than one-way broadcast or memory examination models of education. Immersion education will be greatly considered and learning will not be limited to the classroom, but to actual scenarios where problems are solved using what was learned. Learners enhance the applicability through constant imitation to find problem solutions.

2. The strategy of using AR e-learning will be welcomed by employees in the agricultural technology industry. AR could provide benefits in the learning process, facilitating access to the labor market [21]. An extension of the study findings is that the multi-element AR e-learning platform is favored by users and is the essential facet of the success factor information system–user satisfaction. When employees of the agricultural industry engage in management and investment activities related to information systems, they delve in to fully understand the problem confronted and find a solution through AR e-learning imitation. In particular, before the design of the learning content, drawing on a user-oriented way of thinking, the needs, abilities and constraints of the user have to be well understood. Once all the content has been designed, it is necessary to evaluate the interaction of the products with the learner's operation and control, as well as the function and efficiency of the design output in satisfying the user's expectations. A successful information system is an important condition

for both the industry and its employees in order to meet the ever-variable knowledge economy market. AR experiences have a positive impact on learning [22].

3. Assist the development of agricultural and fishery enterprises in Pescadores and distant districts.

Muhammad Yunus, winner of the 2006 Nobel Peace Prize, set up Grameen Bank in Bangladesh in 1976, dedicating it to providing small loans to the poor. To date, over 5 billion US dollars have been loaned out to local people in Bangladesh, offering weak social group employment opportunities by purchasing products and services from socially vulnerable or marginal racial groups. The innovative microcredit model has since exerted substantial influence all over the world, and was a pioneer of the concept of social enterprise.

To date, social enterprises have developed and extended from the original microcredit concept to address further social issues such as educational opportunities, environment and climate change, etc. Today, there are social entrepreneurs who are improving society through innovative commercial models. These social enterprises, including agricultural enterprises, need much support, not only in terms of professional skills, but also cultural support. Ashoka Foundation, founded in 1981 and dedicated to cultivating social enterprises, has invested tens of millions of US dollars annually, with over 90% of the social enterprises achieving their founding objectives and more than half of them causing changes to national policy, all being due to Ashoka Foundation's strict appraisals and follow-up resource support. Therefore, they may be adopted to assist agricultural and social enterprises in Pescadores and distant districts in the future.

**Author Contributions:** C.-H.L. is responsible for researching purposes and methods; W.-C.W. is responsible for conclusions; C.-Y.L. is responsible for reference and full text integration; P.-N.P. is responsible for Article modification; H.-R.P. is responsible for chart and diagram.

**Funding:** This research received no external funding.

**Acknowledgments:** Zhaoqing University Plan (Guandong, China) and CTBC Business School Plan (Taiwan, China).

**Conflicts of Interest:** The authors declare no conflict of interest.

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
