# Peer review of "Research into the E-Learning Model of Agriculture Technology Companies: Analysis by Deep Learning"

_agronomy, doi:10.3390/agronomy9020083_

Round 1
Reviewer 1 Report
In the introduction, explain the relationship between Data Mining and Deep Learning
At the end of the introduction explain the organization of the article
Explin the relationship between e-learning, augmented reality, deep learning, information system success model, expectation confirmation theory, and theory of communicative action explained in the state of art
To include benefits of the use of RA in educaction. See:
Bacca, J., Baldiris, S., Fabregat, R., Graf, S., Kinshuk, 2014. “Augmented reality trends in education: a systematic review of research and applications. Educ. Technol. Soc. 17 (4), 133-149 http://www.jstor.org/stable/jeductechsoci.17.4.133
Radu, I. “Augmented reality in education: a meta-review and cross-media analysis” Pers Ubiquit Comput (2014) 18: 1533. https://doi.org/10.1007/s00779-013-0747-y
In fiure 3.1 instead of “Theory of Communicative” put “Theory of Communicative Action”
To include questionnarie designed
To explain with more details measurement tool designed (see 332)
In questionnarie designed, there are questions about AR and e-learning?
Author Response
Sorry for the delay. Thank you for your guidance and let us benefit a lot. We have made improvements to the points you have proposed.
Sincerely
Chi-Hsuan Lin
28 December, 2018

Reviewer 2 Report
Excellent study very useful for teachers and employers in the field of agriculture.
Perhaps I would recommend increasing the references in the introduction to compare with similar studies.
Author Response
Sorry for the delay. Thank you for your guidance and let us benefit a lot. We have made improvements to the points you have proposed.
Sincerely
Chi-Hsuan Lin
28 December 2018

Reviewer 3 Report
1.I consider that in the Literature review section the authors should refer to the use of DL and ML in the e-learning sector. They detailed the two concepts but did not refer to articles previously published in this topics(combination between ML, DL and e-learning).
2.Some problems with expressions that need to be changed are listed below:
line 66 - the expression the learner who learned (I suggest to be changed into the student/person who learned)
line 72 - this sentence is nonsense M-Learning is not only e-Learning but also mobile. Should be change or even erased as in the following paragraph M-Learning is explained.
line 100 - as it is not a title it should be more elaborated Two current definitions of Augmented Reality ... are presented below(for example).
Line 397 - 1. AR e-Learning by means of DL Deep Learning carries with successful model. (we suggest that the strikethrough text should be eliminated).
3.Below the tables(4-1,4-2, 4-3 and 4-4) and figures (3-1 and 4-1) it should be specified the source and, in the case of tables, the platform used (for example: Source of information: Author's projection in SPSS/R/RStudio).
The study, especially through the analysis and results part is very interesting and very elaborated. The authors should send it for a check to an English native person for improving the quality of expression. This way, the paper will be more readable and thus the valuable information included in the study will be more significant to the readers.
Author Response

(The authors gave the same response as above.)
